# The Complementary Effects of Dabigatran Etexilate and Exercise Training on the Development and Stability of the Atherosclerotic Lesions in Diabetic ApoE Knockout Mice

**DOI:** 10.3390/ph16101396

**Published:** 2023-10-02

**Authors:** Nikolaos PE Kadoglou, Marianna Stasinopoulou, Evangelia Gkougkoudi, Eirini Christodoulou, Nikolaos Kostomitsopoulos, Georgia Valsami

**Affiliations:** 1Medical School, University of Cyprus, Aglatzia CY 2029, Cyprus; 2Center of Experimental Surgery, Biomedical Research Foundation, Academy of Athens, 11527 Athens, Greece; 3Laboratory of Biopharmaceutics & Pharmacokinetics, Department of Pharmacy, School of Health Sciences, National & Kapodistrian University of Athens, 15771 Athens, Greecevalsami@pharm.uoa.gr (G.V.)

**Keywords:** dabigatran etexilate, exercise training, atherosclerosis, plaque stability, matrix metalloproteinases, inflammation

## Abstract

**Aim:** To determine the complementary effects of dabigatran etexilate (DE), exercise training (ET), and combination (DE + ET) on the development and stability of the atherosclerotic lesions in diabetic apoE knockout (apoE^−/−^) mice. **Methods:** In 48 male apoE^−/−^ diabetic mice, streptozotocin (STZ) was induced for 5 consecutive days. Mice received a high-fat diet (HFD) for 8 weeks and then were randomized into four groups (1. Control/CG, 2. DEG: HFD with DE, 3. ETG: ET on treadmill, 4. DE + ETG: combination DE and ET treatment). At the end of the eighth week, all mice were euthanatized and morphometry of the aortic lesions at the level of aortic valve was obtained. Collagen, elastin, MCP-1, TNF-a, matrix metalloproteinases (MMP-2,-3,-9), and TIMP-1 concentrations within plaques at the aortic valve were determined. **Results:** All active groups had significantly smaller aorta stenosis (DEG:7.9 ± 2.2%, ETG:17.3 ± 5.3%, DE + ETG:7.1 ± 2.7%) compared to CG (23.3 ± 5.5% *p* < 0.05), reduced the relative intra-plaque content of MCP-1, macrophages, MMP-3, and MMP-9, and considerably increased collagen, elastin, and TIMP-1 (*p* < 0.05). Group 4 showed the most pronounced results (*p* < 0.05). Both DEG and DE + ETG significantly reduced MMP-2 and TNF-a concentrations compared to ETG and CG (*p* < 0.010). **Conclusion:** DE and ET treatment of diabetic apoE^−/−^ mice resulted in complementary amelioration of atherosclerotic lesions development and stability, mediated by the anti-inflammatory modulation of both DE and ET.

## 1. Introduction

Atherosclerosis-related complications are by far the leading cause of morbidity and mortality worldwide [1]. Atherosclerotic vulnerable plaques are prone to rupture and contain highly predictive information for future cardiovascular events. Their main characteristics are a large lipid core, thin fibrous cap, abundant inflammatory cell infiltration, inadequate elastin and collagen proteins, and elevated proteolytic enzyme activity, such as matrix metalloproteinases (MMPs) [2]. The formation of atherothrombotic lesions after plaque rupture leads to devastating lumen obstruction. A plethora of studies have been published examining the impact of several pharmaceutical molecules as potential candidates for atherosclerosis inhibition [3,4]. Most importantly, the current preclinical research has shifted to pharmaceutical agents which may not reduce the atherosclerotic burden, but which alter the atherosclerotic plaques’ texture in order to make them less vulnerable and so less prone to rupture [5]. Notably, the co-existence of diabetes precipitates the atherosclerosis progression and magnifies the risk of cardiovascular events [6,7]. Therefore, therapeutic modalities targeting to prevent plaque destabilization and lumen thrombosis are of clinical importance among diabetic patients [5].

As a pleiotropic enzyme, thrombin regulates hemostatic and non-hemostatic functions, including an array of arterial homeostasis and atherosclerosis-related actions [8]. Physiologically, thrombin generation serves mainly to arrest bleeding, and exerts anti-coagulant properties depending on the local condition [9]. In contrast, vascular dysfunction shifts thrombin activities towards adverse thrombogenesis and inflammation. This shift is typically seen under the process of atherosclerosis leading to atherothrombotic complications [10,11]. Thus, the long-term modification thrombin may unravel novel pathways of reducing both the burden of atherosclerosis and its propensity to complications. Dabigatran etexilate (DE) is an oral, reversible, direct competitive thrombin (factor IIa) inhibitor [12]. DE has been shown to be as effective as warfarin in acute VTE treatment and in the secondary prevention of venous thromboembolism (VTE) [13], with reduced major bleeding, and superior (DE 150 mg b.i.d.) to warfarin in non-valvular atrial fibrillation [14]. However, the impact of DE on atherosclerosis-related complications is undetermined based on current data from clinical trials [15,16]. Experimental data in transgenic, atherosclerosis-prone mice have recently supported the beneficial effects of DE on atherosclerosis development and texture [17,18]. The favorable “pleiotropic” modification of inflammatory and oxidative mechanisms has been proposed as the underlying mechanisms, but more data are required.

Robust evidence supports exercise training (ET) as a cornerstone of therapeutic strategies for the primary and secondary prevention of atherosclerosis-induced complications [19]. The beneficial effects of systemic ET on the cardiovascular system are only somewhat explained by the modulation of traditional cardiovascular risk factors (e.g., dyslipidemia, diabetes, etc.) [20], while additional “pleiotropic” actions may also predominantly contribute [21,22]. Multiple anti-atherosclerotic mechanisms, including anti-inflammatory, anti-oxidative, and other pathophysiological pathways have been proposed as the mediators for atherosclerotic lesion suppression or stabilization [23,24]. In clinical setting, atherosclerosis regression is not supported in exercise-trained participants; however, pleiotropic actions of systemic exercise can be beneficial for secondary prevention of atherosclerotic complications [25,26]. Notably, the exercise training exerts “pleiotropic” actions in diabetic animal models which attenuate plaque vulnerability [27]. Similarly, but to a lesser extent, there is indirect evidence of the atheroprotective, “pleiotropic” action of systemic ET on multiple cardiovascular risk factors in the diabetic population [28]. Among exercise-related atheroprotective mechanisms, the attenuation of coagulability has emerged as a promising one in cardiovascular disease-free adults [29]. However, data concerning the impact of exercise training on plaque stability through thrombin modification are absent. 

In the present study, we hypothesized that the combination of the aforementioned therapeutic modalities (DE plus ET) would confer additive results on plaque stability compared to each intervention alone and, thus, their “pleiotropic” anti-inflammatory properties may advocate such a combined treatment.

## 2. Results

The results are listed in Table 1 and Table 2 and Figure 1, Figure 2 and Figure 3. Four mice died before the end of the study (two mice in DE + ETG, one mouse in DEG, one mouse in CG) for unexpected reasons. Those mice were replaced in order to maintain equivalent groups in the statistical analysis. At the end of the study, no significant differences were detected between all groups in the following parameters: body weight, glucose, total cholesterol, and triglycerides levels (*p* > 0.05). Exercise training conferred a slight decline in the latter variables. Fasting plasma glucose (FPG) levels highly increased from baseline to the end of the study as a result of the combination of STZ administration and HFD (*p* < 0.001). Furthermore, the amount of FPG increment was marginally lower in ETG than in DEG (*p* = 0.086) and CG (*p* = 0.092) groups (Table 1).

### 2.1. Mean Plaque Area and Plaque Stability

Results are depicted in Table 2 and Figure 1. Compared to CG, all intervention groups had a significantly smaller mean atherosclerotic plaque area at the aortic valve with a lower percentage of lumen stenosis (*p* < 0.001 and *p* < 0.001, respectively). Notably, both DEG and DE + ETG groups had a significant further reduction in atherosclerotic plaque burden compared to ETG (*p* < 0.001), but they did not differ between them (Table 2 and Figure 1). 

In comparison to CG, the aortic atherosclerotic plaques exhibited a more stable phenotype in all intervention groups. Those stable plaques were characterized by a significantly higher collagen and elastin content and a thicker fibrous cap strengthening the structure of the atherosclerotic lesions against blood flow (*p* < 0.001) (Figure 2). Notably, those effects were most pronounced and significant after combined treatment (DE + ET) rather than each intervention alone (*p* < 0.05). In immunohistochemical analysis, both DEG (*p* = 0.040) and DE + ETG (*p* = 0.023) showed higher concentrations of a-actin antibodies within plaques than CG, while the impact of ET on VSMCs was negligible (Table 2).

### 2.2. Inflammatory Mediators 

In comparison to CG, all three intervention groups appeared with significantly reduced positive-stained atherosclerotic areas at the aortic valve for macrophages (Mac-3 immunostaining, *p* < 0.001), MMP-3 (*p* < 0.05), MMP-9 (*p* < 0.001), and MCP-1 (*p* < 0.001). Notably, the most pronounced reduction in the above parameters was demonstrated in the DE + ETG rather than DEG and ETG groups (*p* < 0.05). Both MMP-2 and TNF-a concentrations were reduced to a similar degree in DEG and DE + ETG compared to ETG and CG (*p* < 0.010), implicating that DE administration predominantly drove the MMP-2 and TNF-a reduction. Inversely, in all intervention groups, we observed increased intra-plaque levels of TIMP-1 compared to CG (*p* < 0.05), while the amount of increment was greatest with combined interventions (DE + ET) compared to ET (*p* = 0.018) or DE (*p* = 0.025) therapy alone. The calculated MMP-9/TIMP-1 ratio was considerably decreased across all intervention groups (*p* < 0.05). The aforementioned results are depicted in Figure 3 and Figure 4.

## 3. Discussion

In our study, all three interventions (DE, ET, DE + ET) considerably suppressed atherosclerotic lesion development, leading to less atherosclerotic burden compared to untreated diabetic mice after an 8-week therapy. Importantly, the stability of the atherosclerotic lesions was improved by all interventions, with the combined treatment (DE + ET) conferring the most pronounced results. The latter was attributed to the complementary anti-inflammatory effects of ET and DE, while alterations in body weight, glucose levels, and lipid profile were negligible across all groups.

As expected, we confirmed our previous data about the suppressive effects of either ET or direct thrombin inhibition on atherosclerosis progression [18,30]. The combination of DE and ET did not further decrease the atherosclerosis burden, presumably due to the exaggerated atherosclerosis regression already achieved after DE administration. Plaque regression after ET was significant, but modest in comparison to our previous studies [17,27]. It is worth mentioning that the aforementioned results were observed despite the absence of any significant change in lipid levels and body weight. Three previously published studies have demonstrated a complex interplay between thrombin and atherosclerosis, using the same animal model of hypercholesterolemic mice [18,19,31]. In those studies, the direct inhibition of thrombin counteracted its pro-inflammatory, oxidative, and pro-atherogenic actions, independent of lipid alterations. In the present study, DE and ET therapy inhibited not only plaque progression but also destabilization by enhancing collagen and elastin contents and fibrous cap thickness, and by suppressing IEL ruptures. It is well-known that MMPs regulate extracellular matrix (ECM) remodeling and inflammatory cells’ infiltration [32], the mainstays of atherosclerosis development [33,34]. Thus, anti-inflammatory therapies targeting the inhibition of MMPs and inflammation have potential anti-atherosclerotic benefits [35]. Atherosclerosis regression after thrombin inhibition has been demonstrated only in animal models [36,37]. Clinical trials assessing atherosclerosis progression have significant disadvantages, like slow atherosclerosis progression, repeated expensive imaging modalities to detect any change in atherosclerosis burden, multifactorial pharmaceutical therapy confounding results [38,39,40]. The experimental atherosclerosis regression is resounding, but its extrapolation to the clinical setting is not feasible. 

Atherosclerotic plaque texture predominantly determines the constant interaction between blood and plaque and the consequent atherothrombotic events [41]. In our study, all the examined parameters that culminated in plaque stability, like elastin, collagen, inflammatory agents (macrophages TNF-a, MCP-1), and MMPs/TIMP-1 [42], were positively affected by each therapeutic modality, producing a more stable phenotype. We and other investigators have demonstrated the strong contribution of inflammation to atherosclerosis vulnerability [43,44]. The balance of proteolytic and anti-proteolytic enzymes (MMPs/TIMPs) has been long studied as a surrogate marker of plaque vulnerability. To our knowledge, this is the first study demonstrating the complementary, favorable changes in most of the aforementioned parameters of plaque stability after combined DE + ET therapy. In particular, the combination of DE and ET increased the intra-plaque concentration of collagen and elastin contents in an additive pattern. That effect was associated with a significant reduction in proteolysis expressed by downregulated MMPs (MMP-2,-3,-9) and increased TIMP-1 levels in all active groups. In addition to this, DE administration rather than ET stimulated the plaque infiltration by VSMCs, the cellular sources of the ECM [45]. Overall, the combined treatment magnified the derived benefits of each intervention (DE and ET) by shifting the balance between production and degradation of ECM in established atherosclerotic lesions towards increased intra-plaque collagen and elastin [46]. The observed complementary effects provide a clinical message of combining pharmaceutical inhibition of thrombin with lifestyle changes, like exercise training. 

We also noticed a complementary pattern after DE and ET in the modulation of inflammatory mediators, with the exception of MMP-2 and TNF-a. Experimental and clinical data have supported the relationship of inflammation with either low physical activity or thrombin action [47,48]. Previous studies have reported a reduction in two multifunctional inflammatory agents, TNF-a [49] and MCP-1 [50], after exercise intervention. The impact of thrombin inhibitors on them is still unknown. In our study, single therapy with either DE or ET inhibited macrophages’ infiltration and decreased macrophage-derived atherogenic cytokines, like MCP-1 and TNF-a, within plaques [51]. The combination therapy seemed to further precipitate the suppression of MCP-1 and TNF-a, and the consequent plaque stabilization, having the highest elastin, collagen, and VSMCs contents and the lowest macrophage concentration. Plaque stabilization has been targeted by many pharmaceutical interventions. The clinical relevance of our preliminary data remains to be proved. 

Macrophages are widely recognized as the cornerstone of atherogenesis and the predominant cellular sources of MMPs [36]. Thus, the observed decrease in macrophage number provides a plausible explanation for the proportional reduction in the protein concentration of MMP-2, MMP-3, and MMP-9 across our groups. The involvement of the latter MMP members in atherosclerotic plaque vulnerability is well-documented [52,53]. On the other hand, TIMP-1 has been recognized as a wide-spectrum MMP inhibitor, and its extensive study has reported its considerable contribution to atherosclerotic plaque stability [54]. Therefore, the alteration in MMPs/TIMP-1 ratio to lower proteolytic activity strongly supports a predominant atheroprotective mechanism. Any intervention which alters this balance in favor of lower MMPs and higher TIMPs activity has been associated with a more stable plaque phenotype and less adverse cardiovascular events [7,55]. Although we have previously demonstrated the beneficial effect of each intervention alone (ET or DE) on MMPs/TIMP-1 equilibrium [12,19], this is the first study where combined therapy confirmed their additive effects on it [18,24]. Those pleiotropic changes after combined intervention are quite promising for optimal ECM homeostasis, but they require unambiguous further validation.

The present study had several drawbacks. First of all, the histochemistry-based measurements provide a relative quantification of molecular concentration, but do not directly express the absolute protein activity and they do not reflect gene expression. Secondly, the estimation of plaque vulnerability is commonly indirect in apoE^−/−^ mice fed HFD. A number of parameters associated with plaque texture are mostly used in this animal model, since plaque rupture is rarely observed in their atherosclerotic lesions. Nevertheless, apoE^−/−^ mouse plaque morphology shows great similarities with that in humans. Third, we observed some unexpected deaths of DE-treated mice, but we were unable to clarify the underlying reason. The detrimental or toxic effects of thrombin inhibition was not the object of the present study.

## 4. Materials and Methods

### 4.1. Study Design

Forty-eight male mice with homozygous deficiency in apolipoprotein-E (apo-E^−/−^, C57BL/6J background), aged 8 weeks, were fed a Western-type high-fat diet (HFD—42% of total calories from milk fat and 0.15% from cholesterol, ( Harlan, Teklad; 88137) for 8 weeks, in order to develop atherosclerotic lesions. Diabetes was induced at the beginning of the study by peritoneal injections of streptozotocin (STZ) for 5 consecutive days (0.05 mg/g body weight in 0.05 mol/L citrate buffer, pH 4.5). Mice maintaining fasting glucose levels >200 mg/dL throughout the course of the study were considered diabetic and were included in analyses. This is a valid animal model for diabetic atherosclerosis development that is widely used in pharmaceutical studies. The advantage of such a model is the combination of hyperglycemia induced with STZ administration due to pancreas devastation and reduced insulin secretion and most of the metabolic syndrome characteristics induced by HFD. Thereby, diabetic mice combine hypercholesterolemia and adipose tissue accumulation. Before beginning the study, mice were randomized into the following equivalent groups (n = 12): (1) CG: no intervention was performed, and the mice of this group served as controls. (2) DEG: HFD was supplemented mixed with DE (10 mg DE/g chow) for the whole study period. (3) ETG: mice underwent ET on a treadmill (Exer-6M Open Treadmill, Columbus Instruments, Columbus, Ohio USA). Mice were gradually accustomed to the training program. In particular, we gradually increased the duration of each session from the first week (15 min) to the second week (45 min), with a 2-minute rest interval at the medium of the session. Each week, the mice performed a 5-day ET program in order to maintain cardiovascular adaptations. During the first 2 weeks, the speed of the treadmill was increased from 8 m/min to 12 m/min. After the second week, the time of each session was further increased to 60 min, keeping an intermediate 2-minute rest interval and, in parallel, the speed approached 15 m/min. The slope of the treadmill was kept unchanged at 5° throughout the training period. The following exercise parameters (5 times/week, 60 min/session, velocity of 12 m/min, slope of 5°) remained constant until the end of the 8th exercise week. Some mechanical and electrical shock-plate stimuli were used during the first session on the treadmill, and this was required to different degrees in individual mice [12]. After that session, mechanical stimuli were provided occasionally by tapping very gently on the back of the mice with a wooden stick. (4) DE + ETG: combination of DE and ET treatment. Mice received the mixed diet (HFD + DE) and during the same period underwent the ET program, as described previously. All mice were euthanized at the end of the study (8th week) under isoflurane deep anesthesia. Diets were provided by Boehringer Ingelheim, Biberach, Germany. As we have previously reported, the daily absorbance of active substance was calculated to be ~2.3 mg of DE [18]. Just before euthanasia, body weights were measured, and blood samples were obtained through gastrocnemius muscle puncture. 

All the experiments were performed with prior approval by the Ethics Committee for Animal Experimentation of the Biomedical Research Foundation, Academy of Athens, and the competent Veterinary Service, according to the European Directive 2010/63 (permit number: 43029201). All surgeries were performed under isoflurane anesthesia to minimize suffering. The animals were kept in animal rooms in a 12 h light/12 h dark environment. Light started at 07:00 a.m., using a light density of 300 Lux, positioned 1 m above the floor in the middle of the room. Specific pathogen-free (SPF) conditions were maintained for all animals, and they were housed at a room temperature (22.1 °C), with 55 ± 10% relative humidity. Animal rooms were operated with a positive air of 0.6 Pa. Tap water in drinking bottles and vacuum-packed pelleted food were provided ad libitum. The mice were handled in accordance with the relevant international guidelines for the proper care and use of laboratory animals.

### 4.2. Glucose Tolerance Test

We selected 6–7 mice per group one week prior to euthanasia. Those mice underwent an intraperitoneal glucose tolerance test (GTT) in anesthetized mice. After an overnight fasting, glucose (D-glucose 50% wt/vol solution) was injected intra-peritoneally via a 27-gauge needle. The dose was set at 2 g/kg of body weight. Blood samples for glucose measurements were taken before glucose injection and at 30, 60, 90, and 120 min after injection. Blood glucose was measured at each time point by Accu-Chek advantage glucose monitors (Roche Diagnostics, Indianapolis, IN, USA), and the area under the curve (AUC) was then determined using the trapezoid rule.

### 4.3. Histology

After euthanasia, we punctured the heart and through that we perfused the heart along with the aortic root by injecting normal saline. After excision of the aortic root, we fixed it in 10% buffered formalin overnight and then we embedded it in paraffin blocks. The procedure of histomorphometric analysis has been previously described [56]. In a microtome, serial sectioning (5 μm thickness) started when aortic valve leaflets become apparent. For each staining per mouse, we mounted on poly-D-lysine-coated slides 3 nonconsecutive aortic slices (the intervals were kept equal at 20 μm aortic root length) following a standardized protocol. That was necessary in order to colocalize the various measured variables and get comparable results. For quantitative morphometric analysis, we stained cross-sections with hematoxylin/eosin (H&E), sirius red and orcein, according to a standardized protocol [57]. The former staining was used for morphometric analysis of atherosclerotic burden. Finally, by means of immunohistochemistry, we stained paraffin sections of the aortic valve using antibodies directed against the corresponding antigens: monocytes chemoattractant protein-1 (MCP-1), tumor necrosis factor-a (TNF-a), MMP-2, MMP-3, MMP-9 (MBL, International Corporation, Woburn, MA, USA), Mac-3 antigen of murine macrophages (BD Pharmigen, Franklin Lakes, NJ, USA), alpha-smooth muscle isoform of actin (Biocare Medical, LLC, Concord, CA, USA) and TIMP-1 (Acris Antibodies GmbH, Herford, Germany). 

### 4.4. Digital Processing—Histomorphometry

A bright-field microscope (Leica DM LS2, Wetzlar, Germany) was used to observe all sections from the aortic valves, by magnifying ×10 the acquired digital pictures. All of them were stored in a lossless format using a Leica DC 500 microscope camera (Heerbrugg, Switzerland) and the “Altra 20 Soft Imaging System” computer software(APX 100). In H&E-stained sections, we measured the extent of the atherosclerotic plaques (in μm^2^) and the total lumen area (in μm^2^) circumscribed by the internal elastic lamina (IEL). The percentage of luminal stenosis was calculated by dividing the total atherosclerotic plaque area by the total lumen area in each section. For the quantification of lesions in aortic sections per animal, we averaged the plaque area and the luminal stenosis. In Sirius red and orcein-stained sections, we measured the relative concentrations of collagen and elastin, respectively, as well as the fibrous cap thickness of each atherosclerotic plaque including possible ruptures (i.e., discontinuities or fractures) of the IEL. We then averaged the values of all plaques per animal. For the measurement of the relative concentrations of the stained molecules by immunohistochemistry, the segmental stained plaque area was expressed as the percentage of the whole atherosclerotic plaque area [18]. We then averaged the values of all plaques per animal and then for each group.

### 4.5. Blood Analyses

After an overnight fasting, we obtained all blood samples via gastrocnemius muscle puncture under isoflurane anesthesia. Fasting glucose, triglycerides, and total cholesterol plasma levels were immediately assayed in an automatic enzymatic analyzer (Olympus AU560, Hamburg, Germany).

### 4.6. Statistical Analysis

Our results were presented as mean values ± standard deviation. Distribution normality was assessed using the Shapiro-Wilk test. For continuous variables, the comparisons within and between groups was made using paired samples *t*-test and one-way ANOVA and post hoc Tukey test, respectively. Within groups analysis was performed only for the changes of variables from baseline to the end of the study. We conformed to published suggestions for medicinal chemistry modelling. Statistically significant difference was considered when a two-tailed *p* value < 0.05 was found. All statistical analyses were based on SPSS v28.0 (IBM, Armonk, NY, USA).

## 5. Conclusions

In conclusion, combined DE and ET treatment in diabetic atherosclerotic apoE^−/−^ mice showed additive beneficial effects on most parameters of atherosclerotic lesion stability. Combined treatment yielded a more stable plaque phenotype, which theoretically would have reduced the risk of rupture. Those effects paralleled a favorable modification of inflammatory mediators, implicating a strong impact of both DE and ET on those crucial parameters of atherosclerosis progression. Our findings outline the clinical importance of pharmaceutical intervention accompanied with lifestyle alteration in the management of atherosclerotic complications.

## Figures and Tables

**Figure 1 pharmaceuticals-16-01396-f001:**
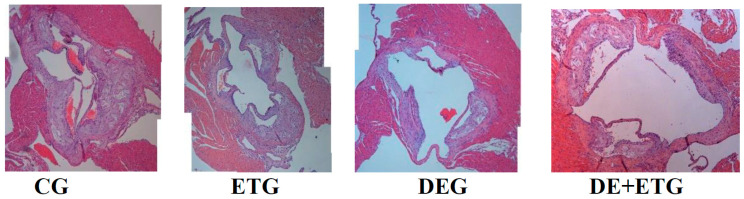
All active groups (dabigatran etexilate, exercise training, combined treatment) significantly reduced plaque formation compared to controls in ApoE^−/−^ mice. Representative images and quantifications of aortic valve sections stained with hematoxylin/eosin, across all groups.

**Figure 2 pharmaceuticals-16-01396-f002:**
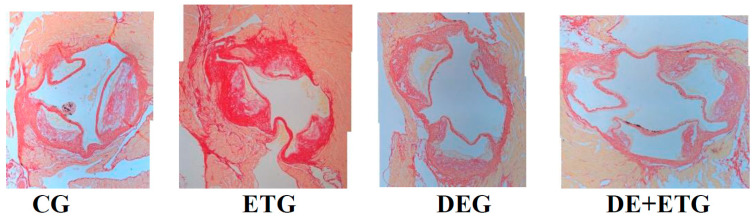
All active groups (dabigatran etexilate, exercise training, combined treatment) significantly enhanced collagen concentrations compared to controls in ApoE^−/−^ mice. Representative images of aortic valve sections stained with Sirius red (collagen) across all groups.

**Figure 3 pharmaceuticals-16-01396-f003:**
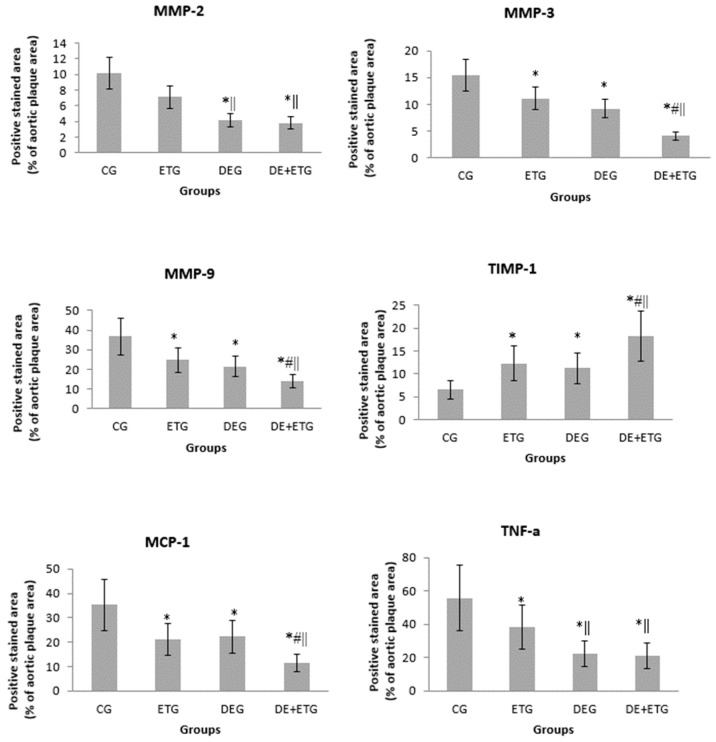
Quantification of immunohistochemical staining with antibodies against MMP-2, MMP-3, MMP-9, TIMP-1, TNF-a, and MCP-1. * *p* < 0.05 vs. CG, # *p* < 0.05 vs. DEG, ‖ *p* < 0.05 vs. ETG.

**Figure 4 pharmaceuticals-16-01396-f004:**
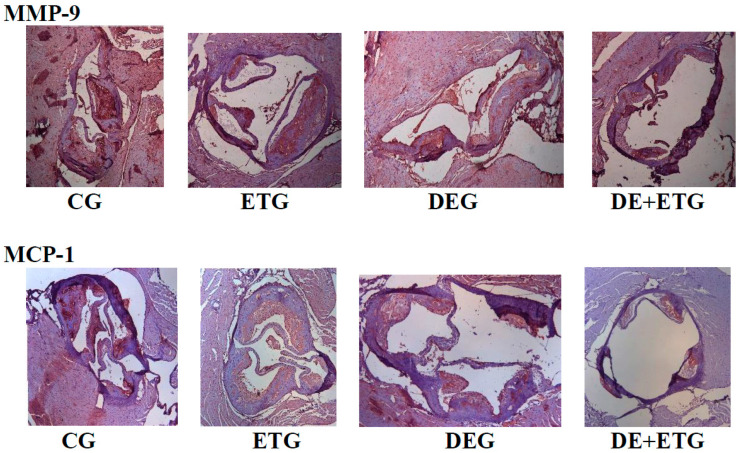
Representative images of plaques at aortic valve sections stained with MMP-9 and MCP-1 across all groups. Section thickness was set at 5 μm and the original magnification was 100×. We analyzed 2–3 sections of each mouse (6 mice per group). CG, control group; DEG, dabigatran etexilate group; ETG, exercise training group; DE + ETG, dabigatran etexilate + exercise training group.

**Table 1 pharmaceuticals-16-01396-t001:** Body weight, fasting plasma glucose (FPG), total cholesterol, and triglycerides levels at baseline and at the end (8 weeks) of the study.

	CG (n = 12)	DEG (n = 12)	ETG (n = 12)	DE + ETG (n = 12)	*p*
**Weight (g)**					
**Baseline**	27.8 ± 4.6	28.5 ± 7	28.3 ± 5.1	27.2 ± 4	0.881
**End**	31.1 ± 4.1	32.23 ± 6.3	29.8 ± 2.7	29.7 ± 5	0.321
**FPG (mg/dL)**					
**Baseline**	113 ± 28	108 ± 31	103 ± 21	120 ± 41	0.773
**End**	165 ± 44 *	189 ± 45 *	165 ± 52 *	182 ± 49*	0.654
**TC (mg/dL)**					
**Baseline**	358 ± 145	398 ± 103	404 ± 198	387 ± 134	0.298
**End**	684 ± 169 *	612 ± 223 *	556 ± 110 *	579 ± 311 *	0.112
**TG (mg/dL)**					
**Baseline**	82 ± 13	84 ± 21	89 ± 28	94 ± 31	0.498
**End**	125 ± 29 *	115 ± 24 *	126 ± 31 *	116 ± 32 *	0.687

CG, control group; DEG, dabigatran etexilate group; ETG, exercise training group; DE + ETG, dabigatran etexilate + exercise training group. TC, total cholesterol; TG, triglycerides; P, one-way ANOVA. ** p* < 0.05, within groups.

**Table 2 pharmaceuticals-16-01396-t002:** Atherosclerotic plaque area, lumen area, and percentage of lumen stenosis (H&E staining), intra-plaque contents of elastin (orcein staining), collagen (Sirius red staining), vascular smooth muscle cells (VSMCs—a-actin) and macrophages (Mac-3).

	CG (n = 12)	DEG (n = 12)	ETG (n = 12)	DE + ETG (n = 12)	** *p* **
**Plaque area (×10^3^ μm²)**	287.9 ± 54.12	105.12 ± 31.11 ^a,c^	201.65 ± 65.12 ^a,b,d^	72.23 ± 15.51 ^a,c^	<0.001
**Lumen area (×10^3^ μm²)**	1323.56 ± 265.3	1355.32 ± 288.54	1222.1 ± 243.51	1274.76 ± 289.23	0.885
**Lumen stenosis (%)**	23.3 ± 5.5	7.9 ± 2.2 ^a,c^	17.3 ± 5.3 ^a,b,d^	7.1 ± 2.7 ^a,c^	<0.001
**Elastin (%) plaque**	9.79 ± 2.92	21.62 ± 6.52 ^a,d^	18.91 ± 5.07 ^a,d^	30.24 ± 7.72 ^a,b,c^	0.002
**Collagen (%) plaque**	16.45 ± 8.08	26.83 ± 4.79 ^a,d^	21.44 ± 3.1 ^a,d^	31.41 ± 4.88 ^a,b,c^	0.001
**Fibrous cap thickness (μm)**	12.52 ± 2.18	22.32 ± 3.46 ^a,d^	19.63 ± 3.02 ^a,d^	31.41 ± 4.12 ^a,b,c^	<0.001
**a-actin (VSMCs) (%) plaque**	15.91 ± 4.98	23.79 ± 5.54 ^a^	18.51 ± 4.31 ^a,d^	25.22 ± 6.18 ^a,c^	0.039
**Mac-3 (macrophages) (%) plaque**	28.85 ± 9.52	19.46 ± 5.58 ^a,d^	21.02 ± 4.93 ^a,d^	12.33 ± 2.87 ^a,b,c^	<0.001

CG, control group; DEG, dabigatran etexilate group; ETG, exercise training group; DE + ETG, dabigatran etexilate + exercise training group. P, one-way ANOVA *p* value. Significant differences of each intervention group (*p* < 0.05) compared to other groups based in a post hoc one-way ANOVA analysis: ^a^ CG, ^b^ DEG, ^c^ ETG, ^d^ DE + ETG.

## Data Availability

The data presented in this study are available in the present article.

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
