# Peer review of "The Complementary Effects of Dabigatran Etexilate and Exercise Training on the Development and Stability of the Atherosclerotic Lesions in Diabetic ApoE Knockout Mice"

_pharmaceuticals, 2023, doi:10.3390/ph16101396_

Round 1

Reviewer 1 Report

Dear Editor,

I read with interest the review written by Kadoglou et al., in which they analyze the impact of combined therapy with Debigatran Etexilate and Exercise Training on plaque stability on an animal model for diabetic atherosclerosis (apo-E-/-, C57BL/6J background). Additionally, they compared this therapy with each single treatment. Following the study, the authors demonstrated that DE and ET treatment in diabetic apoE-/- mice showed complementary amelioration of atherosclerotic lesions development and stability, mediated by the anti-inflammatory modulation.

The article is current, very well structured and easy to write. The authors enrolled a consistent number of 48 children in this study.

I recommend the authors to improve the quality of the Figures, as well as to join figures 1a and 1b or to present them as separate figures.

In the future, it would be very interesting if the authors would also introduce statin treatment to see what impact DE + Statin + ET would have

Author Response

I read with interest the review written by Kadoglou et al., in which they analyze the impact of combined therapy with Debigatran Etexilate and Exercise Training on plaque stability on an animal model for diabetic atherosclerosis (apo-E-/-, C57BL/6J background). Additionally, they compared this therapy with each single treatment. Following the study, the authors demonstrated that DE and ET treatment in diabetic apoE-/- mice showed complementary amelioration of atherosclerotic lesions development and stability, mediated by the anti-inflammatory modulation.

 The article is current, very well structured and easy to write. The authors enrolled a consistent number of 48 children in this study.

 I recommend the authors to improve the quality of the Figures, as well as to join figures 1a and 1b or to present them as separate figures.

We have tried to improve the quality, but we did not succeed as much as we would like. This is due to the insertion in the manuscript. We have presented figures 1a and 1b separately as figure 1 and 2. 

In the future, it would be very interesting if the authors would also introduce statin treatment to see what impact DE + Statin + ET would have

Thank you for your suggestion and we are currently working on that

Reviewer 2 Report

After careful revision of the manuscript it was found that the manuscript covers an important topic and worthy of study and can be published after minor revision which does not affect the quality of the manuscript.

-         Considering the statistical data, please find bellow some suggestions to improve the quality of the manuscript.

1)           The authors should report applicability domain of the developed models according to Chemom Intell Lab Sys, 145, 2015, 22-29, http://dx.doi.org/10.1016/j.chemolab.2015.04.013.

2)           The authors should use rm2 metrics for validation. See J Comput Chem 34, 2013, 1071-1082 and Journal of Chemistry, v. 2016, p. 1-12, 2016 (http://dx.doi.org/10.1155/2016/9198582). All suggested references should be included in the paper as well.

Author Response

After careful revision of the manuscript it was found that the manuscript covers an important topic and worthy of study and can be published after minor revision which does not affect the quality of the manuscript.

-         Considering the statistical data, please find bellow some suggestions to improve the quality of the manuscript.

 1)           The authors should report applicability domain of the developed models according to Chemom Intell Lab Sys, 145, 2015, 22-29, http://dx.doi.org/10.1016/j.chemolab.2015.04.013.

 2)           The authors should use rm2 metrics for validation. See J Comput Chem 34, 2013, 1071-1082 and Journal of Chemistry, v. 2016, p. 1-12, 2016 (http://dx.doi.org/10.1155/2016/9198582). All suggested references should be included in the paper as well.

We really appreciate the reviewer’s suggestion. We have already added the first proposed reference. However, we cannot find any convenient place to add the second proposed reference, since validation has been done by the supplier and it was not necessary for our study.

Reviewer 3 Report

The manuscript presents modern research devoted to the study of the complementary effects of dabigatran etexilate and exercise training on the development and stability of atherosclerotic lesions in diabetic ApoE knockout mice. The results obtained by the authors are of interest for developing future research in this direction. The results are obtained based on well-planned and designed experiments and are clearly presented. I believe that the work fits and can be published in the special issue.

Author Response

The manuscript presents modern research devoted to the study of the complementary effects of dabigatran etexilate and exercise training on the development and stability of atherosclerotic lesions in diabetic ApoE knockout mice. The results obtained by the authors are of interest for developing future research in this direction. The results are obtained based on well-planned and designed experiments and are clearly presented. I believe that the work fits and can be published in the special issue.

Thank you very much for your kind comments